# ADAPTIVE PSEUDO-LABELING FOR QUANTUM CALCULATIONS

## ABSTRACT

Machine learning models have recently shown promise in predicting molecular quantum chemical properties. However, the path to real-life adoption requires (1) learning under low-resource constraints and (2) out-of-distribution generalization to unseen, structurally diverse molecules. We observe that these two challenges can be alleviated via abundant labels, often not the case in quantum chemistry. We hypothesize that pseudo-labeling on a vast array of unlabeled molecules can serve as gold-label proxies to expand the training labeled dataset significantly. The challenge in pseudo-labeling is to prevent the bad pseudo-labels from biasing the model. Motivated by entropy minimization framework, we develop a simple and effective strategy PSEUD$\sigma$ that can assign pseudo-labels, detect bad pseudo-labels through evidential uncertainty, and prevent them from biasing the model using adaptive weighting. Empirically, PSEUD$\sigma$ improves quantum calculations accuracy across full data, low data, and out-of-distribution settings.

## 1 INTRODUCTION

Accurate quantum mechanical (QM) calculations of drug-like molecules at CCSDT (coupled cluster single-double-triple) or MP2 (second order Møller-Plesset) (Watts et al., 1992) level of theory, which is essential to characterize biomolecular interactions, continue to be prohibitively expensive, despite recent advances in hardware capabilities. Machine learning (ML) models have astonishing performance in approximating these calculations at a fraction of the computational cost (von Lilienfeld & Burke, 2020). Such speedups could accelerate the discovery of new therapeutics.

In the absence of large-scale benchmark data sets reporting CCSDT or MP2 level calculations, most publications on this topic have relied on QM9, a standard benchmark of Density Functional Theory (DFT)-level energy and properties, for training and evaluating QM/ML models. This data set contains molecular geometries and energies that were generated using DFT, which provides a faster (compared to CCSDT, for example), but less accurate description of molecular energetics and properties. Previous publications have utilized computed geometries and properties from in-distribution molecules in QM9 to test and validate model performance, showing excellent performance of QM/ML models as demonstrated by low mean absolute error (MAE) of predicted energies of other properties relative to the gold-standard DFT calculations (i.e., gold label). However, it is unclear how the reported architectures (e.g., SchNet (Schütt et al., 2017)) would perform in the regime of low but accurate data (CCSDT or MP2).

Two challenges remain in the way of the realistic adoption of ML-aided QM calculations. Firstly, training molecules only cover part of the distribution, and real-world adoption requires out-of-distribution generalization. Secondly, computing QM properties for datasets of the size of QM9 is costly. Widespread applicability of ML for QM (e.g., such as to CCSDT/MP2) thus requires models to generalize well given low-data constraints. Both challenges listed above can be attributed to the difficulty and cost of generating experimental data in chemistry and materials science. If we have a large and diverse set of labeled molecules, the ML model can generalize to larger chemical spaces and achieve better predictive performance.

**Present work.** Our key observation is that state-of-the-art ML models can obtain a low-precision estimation of the properties even given a small set of labeled molecules. This suggests that the ML predicted label for any unlabelled molecule can be used as a low-precision estimation of the true label. We hypothesize that by considering these low-precision pseudo-labels as the training

targets in the training process, we can largely increase the training dataset size and diversity and alleviate the fundamental label scarcity issue, along with the associated challenges in low-data and out-of-distribution generalization.

In our study, we develop a simple, effective, and model-agnostic pseudo-labeling strategy called PSEUD$\sigma$. Notably, we solicit an extensive array of unlabeled molecules from the PC9 dataset (Glavatskikh et al., 2019) where an ML model assigns pseudo-labels for them. This generates a large set of "labeled" training data points, even when only a relatively small number of reference QM properties are available for training. As the unlabeled dataset is diverse and unseen in QM9, it also helps generalization to unseen data. One crucial issue in pseudo-labeling is the introduced bias from low-quality pseudo-labels. To resolve this, we rely on a key observation that a data point with less evidence/higher model uncertainty is more likely to be of low-quality pseudo-label (Section 6). Thus, we use model-generated evidential uncertainty to quantify each unlabeled data and use it to adaptively lower the weight of bad pseudo-labels in the training loss to reduce the bias effect.

In summary, we have made the following contributions: (1) Previous QM focus on in-distribution and label abundant setting while we point out that the realistic ML-aided QM challenges lie in low-data and out-of-distribution settings; (2) Pivoting away from the status quo in improving the physics-based representation, we propose to look at data-centric approaches on learning from the vast array of unlabeled molecules; (3) We propose an episodic pseudo-labeling scheduling strategy designed specifically for QM since many classic pseudo-labeling tricks do not work well in QM (e.g., data augmentation, model noise, student-training, re-initialization); (4) To prevent bad pseudo-label in biasing the training, we devise an adaptive weighting scheme where the weights are generated using evidential uncertainty such that bad pseudo-labels are automatically filtered out; (5) We derive a theoretical connection between the evidential loss and entropy minimization framework; (6) Empirically, we show PSEUD$\sigma$ can improve QM accuracy for any atomistic model across full-data, low-data, and out-of-distribution settings.

## 2 RELATED WORKS

**ML-aided quantum calculations.** Recently, many ML models have been proposed to improve quantum calculations. They mainly focus on improving the physics-based representation in the full QM9 dataset setting (Schütt et al., 2017; Unke & Meuwly, 2019; Anderson et al., 2019; Lu et al., 2019; Klicpera et al., 2020; Liu et al., 2021; Qiao et al., 2021). In contrast, our work proposes to shift the focus on improving training strategy instead of the model architecture. In addition, PSEUD$\sigma$ is model-agnostic, and it can improve on any atomistic model. Additionally, we focus on realistic quantum calculations such as learning in the low-data regime and out-of-distribution inference.

**Pseudo-labeling.** Pseudo-labeling/self-training generates pseudo-labels for unlabeled data. Numerous works exist for how to assign pseudo-labels, notably through trained ML model prediction (Lee et al., 2013), label propagation(Shi et al., 2018; Iscen et al., 2019), and history cache (Likhomanenko et al., 2021; Higuchi et al., 2021). PSEUD$\sigma$ is different as it focuses on detecting and preventing bad pseudo-label from affecting the model. Also, PSEUD$\sigma$ adopts a novel episodic pseudo-label strategy with a re-initialized learning rate. (Xie et al., 2020) re-initialize the network as a student when a new pseudo-label set is generated along with noise per epoch. In contrast, PSEUD$\sigma$ has no student, and no noise as both are shown to be ineffective for QM in Section 6. In addition, small perturbational noise in 3D molecular geometry could easily lead to a drastic energy difference. Thus, the noise strategy does not work for QM tasks. More related is a concurrent work (Rizve et al., 2021) that develops an uncertainty-aware pseudo-labeling strategy, but they introduce additional hyperparameters to remove pseudo-labels at some uncertainties. In contrast, PSEUD$\sigma$ uses an effective adaptive weighting scheme, along with an episodic pseudo-labeling training schedule. Additionally, PSEUD$\sigma$ is the first method that studies pseudo-label in quantum calculations, which present unique challenges and motivations.

**Uncertainty.** Model uncertainty is a well-studied subject (Kendall & Gal, 2017; Lakshminarayanan et al., 2017; Blundell et al., 2015). (Amini et al., 2020) use evidential uncertainty to add a prior over the gaussian parameters to search for higher-order patterns for regression tasks. PSEUD$\sigma$ leverages evidential uncertainty as the uncertainty measure. Note that PSEUD$\sigma$ is uncertainty measure-agnostic. We can easily switch to alternative uncertainty measures. Recently, (Soleimany et al., 2021) adapt evidential uncertainty and has shown it can successfully help guide property prediction. In contrast,

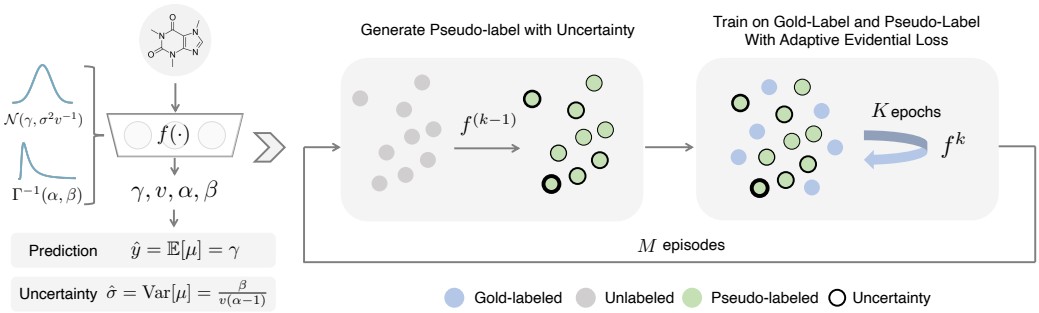

Figure 1: PSEUD$\sigma$ illustration. In every episode $k$, PSEUD$\sigma$ assigns pseudo-labels along with their evidential uncertainty using trained neural network $f^{(k-1)}$ from previous episode. The uncertainty is used as weight to adaptively adjust the loss in this episode's neural network $f^{(k)}$'s training to reduce the effect of bad pseudo-labels in an inner-loop training with $N$ epochs.

we leverage evidential uncertainty as a proxy for pseudo-label quality to tackle low-data and out-of-distribution challenges in realistic quantum calculations setup.

## 3 PROBLEM FORMULATION

Let $\mathcal{X} = \{\mathbf{x}_1, \dots, \mathbf{x}_N\}$ denote $N$ molecules, where each molecule $\mathbf{x}_i$ is uniquely defined by 3D coordinates $\{(a_j^i, b_j^i, c_j^i)\}_{j=1}^{N_i}$ for $N_i$ atoms with atom types $\{t_j\}_{j=1}^{N_i}$ in the corresponding molecule. We then denote $\mathcal{Y} = \{y_1, \dots y_N\}$ a set of quantum mechanical properties for each molecule. The labeled dataset thus consists of a set of pairs of 3D coordinates and scalar labels $\mathcal{D} = \{\mathcal{X}, \mathcal{Y}\}$.

In addition to the labeled data, we solicit a large quantity of unlabeled data to generate pseudo-labels. We denote an unlabeled dataset $\mathcal{U} = \{\mathbf{x}_1, \dots, \mathbf{x}_M\}$, where $M$ is the size of the unlabeled dataset. Given an atomistic model $f(\cdot)$, we can generate pseudo-labels $\hat{\mathcal{Y}} = \{\hat{y}_1, \dots, \hat{y}_M\}$, where $\hat{y}_i = f(\mathbf{x}_i)$ for $\mathbf{x}_i \in \mathcal{U}$.

The problem is to find a machine learning-based atomistic model $f : \mathbf{x} \mapsto y$ that can establish an accurate map from 3D coordinates to the quantum mechanical properties of the molecules, with the help of pseudo-labeled dataset $\mathcal{U}$.

## 4 PSEUD$\sigma$: ADAPTIVE PSEUDO-LABELING FOR QUANTUM CALCULATIONS

PSEUD$\sigma$ (Figure 1) is an approach for quantum chemical property prediction. Building on theoretical motivation from Section 5, PSEUD$\sigma$ solicits pseudo-labels on a vast array of an unlabeled dataset to increase the diversity of the training space via an episodic labeling strategy. Then, it adaptively weights the pseudo-labels using evidential uncertainty to allow a positive transfer. The overview is in Algorithm 1.

**Episodic Pseudo-labeling.** We devise a pseudo-labeling strategy that can ensure learning from the pseudo-labels to the fullest extent for QM. We have made two distinct modifications compared to existing works. First is the pseudo-label scheduling. In the standard pseudo-labeling (Lee et al., 2013), pseudo-labels are updated in every update and the model is continuously trained. In contrast, we devise an episodic training strategy, where each episode consists of $K$ epochs, and pseudo-labels are regenerated in every episode, while the model is continuously trained. This is important because we observe that fast updates on pseudo-labeling prevents the model to fully learn from all useful information in pseudo-labels. In contrast, our episodic approach gives the model more time to absorb useful information from a given set of pseudo-labels. Second modification is the model update. In self-training (Rizve et al., 2021; Xie et al., 2020), a set of pseudo-labels are regenerated after $K$ epochs and the model is reinitialized. Instead, we train the same model throughout episodes. This new strategy allows the model to be exposed to a larger number of labels or training data points given the same time frame. For each episode, we also reinitialize the learning rate with a small step-wise

decay strategy to allow the model a chance to jump out of the local optimum from the previous set of pseudo-labels.

Formally, PSEUD$\sigma$ mainly consists of three stages: in the first stage, regular training is conducted on labeled data $D$, and the output model is the initialized model $f^{(1)}$. In the second stage, the updated model at episode $k$ then conducts inference on the entire unlabeled data $\hat{\mathcal{Y}} = f^{(k)}(\mathcal{U})$ to generate the pseudo-label set. The per-episode pseudo-label set is then combined with the gold-labeled data to form the training data for the next episode. In the third stage, the model is further trained using the combined dataset to get a new model $f^{(k+1)}$ after $N$ epochs. The second and third stages are then reiterated till the loss stops decreasing.

**Evidential uncertainty quantification.** Pseudo-labels are noisy. Many are incorrect and can potentially lead to negative transfer. Thus, it is instrumental in detecting the quality of pseudo-labels. However, there is no auxiliary information in the dataset about the pseudo-labels. Thus, we need to quantify it through some proxies that can be assigned without auxiliary information. Our key observation is that low-quality pseudo-labels have high model uncertainty, and high-quality pseudo-labels have low model uncertainty. Another advantage of model uncertainty is that it can be estimated solely from $\mathbf{x}$, if we make it model uncertainty-aware.

Building on the theoretical motivation about the connection between evidential uncertainty and entropy minimization in Section 5, we use evidential uncertainty as to the proxy for label quality. The evidential modeling of molecular property allows us to derive analytical solution of uncertainty, which could be directly used to weight the pseudo-labels. Formally, we can model the label probabilistically as it is drawn from $(y_1, \cdots, y_i) \sim \mathcal{N}(\mu, \sigma^2)$, where the mean $\mu$ are variance $\sigma^2$ are unknown. To estimate them, we pose a prior

$$\mu \sim \mathcal{N}(\gamma,\, \sigma^2 v^{-1}), \sigma^2 \sim \Gamma^{-1}(\alpha, \beta), \tag{1}$$

where the parameters $\theta = (\mu, \sigma)$ is an instantiation of the posterior $p(\mu, \sigma^2 | \gamma, v, \alpha, \beta)$. The choice of prior allows the factorization $p(\mu, \sigma^2) = p(\mu)\, p(\sigma^2)$ (Jordan, 2009). The posterior then becomes a $\mathrm{NormalInvGamma}(\gamma, v, \alpha, \beta)$ where the maximum likelihood estimation of $\theta$ can be analytically found as

$$\mathbb{E}[\mu] = \gamma,\, \mathbb{E}[\sigma^2] = \frac{\beta}{\alpha - 1}. \tag{2}$$

Here, $\mathbb{E}[\sigma^2]$ plays the role of the aleatoric (data) uncertainty. The uncertainty of the model prediction can also be calculated, i.e. epistemic uncertainty:

$$\mathrm{Var}[\mu] = \mathbb{E}[\sigma^2]/v = \frac{\beta}{v(\alpha - 1)}. \tag{3}$$

As the MLE is deterministic, the model can output the four prior parameters $\{\gamma, v, \alpha, \beta\}$ directly where the prediction and uncertainty can be derived from them analytically. The prior is optimized by evidential loss $\mathcal{L}^{\mathrm{evi}}$ (Amini et al., 2020):

$$\mathcal{L}_i^{\mathrm{evi}} = -\log \mathrm{St}\left(y_i; \gamma, \frac{\beta(1 + v)}{v\alpha}, 2\alpha\right) + \lambda|y_i - \gamma|(2v + \alpha), \tag{4}$$

where the first term is to maximize the log-likelihood of the posterior predictive, which is derived as the Student's t-distribution. The second term is a regularizer that encourages lower total evidence, which correlates positively to higher epistemic uncertainty, when the gap between ground truth and prediction is high (i.e. when model prediction is with high error). Similarly, it encourages lower uncertainty when the model prediction is errorneous. This encourages the model to generate an accurate estimate of uncertainty or the degree of errors for the pseudo-labeled data points. The regularization is controlled by a hyperparameter $\lambda$.

**Adaptive weighting.** The evidential uncertainty detects the low-quality pseudo-labels. The next step is to remove the noisy effect from the model training. Naive methods often use removal based on a threshold (Rizve et al., 2021). However, it has two disadvantages: (1) it introduces a new hyperparameter - the threshold; (2) it removes a portion of unlabeled noisy data, which can contain useful information. Instead, we propose an adaptive weighting mechanism that adapts the evidential loss given the inverse epistemic uncertainty. Intuitively, a higher uncertainty data point requires a lower effect in the loss function because it is more likely that the sample has a low pseudo-label quality,

---

**Algorithm 1:** PSEUD$\sigma$ Algorithm.

---

**Input**: Labeled data $\mathcal{D} = \{(\mathbf{x}_1, y_1), \cdots, (\mathbf{x}_N, y_N)\}$, unlabled data $\mathcal{U} = \{\mathbf{x}_1, \cdots, \mathbf{x}_{N_U}\}$

$\hat{\mathcal{U}} \leftarrow \{\}, \hat{\mathcal{W}} \leftarrow \{\}$          // Initialize with empty unlabeled data

**for** $k \in \{1, \cdots, K\}$          // Outer-loop with $K$ episodes

**do**

     $\mathcal{T} \leftarrow \mathcal{D} \cup \hat{\mathcal{U}}$          // Join updated pseudo-labels

     **for** $(\mathbf{x}_i, \mathbf{y}_i) \in \mathcal{T}$          // Inner-loop with $N$ epochs

     **do**

         $\theta_i = (\gamma_i, v_i, \alpha_i, \beta_i) = f^{(k-1)}(\mathbf{x}_i)$      // Evidental parameters

         $\hat{\mathbf{y}}_i = \mathbb{E}[\mu] = \gamma_i$      // Posterior prediction

         $\mathcal{L} = \mathrm{L}(\hat{\mathbf{y}}_i, \mathbf{y}_i, \theta_i, \hat{\mathcal{W}}_i)$     // Adaptive evidential loss via Eq. 5

         $f^{(k-1)} = \mathrm{Update}(f^{(k-1)}, \mathcal{L})$      // Inner-loop update

     **end**

     $f^{(k)} \leftarrow f^{(k-1)}$      // Update teacher model for pseudo-labels

     **for** $\mathbf{x}_i \in \mathcal{U}$ **do**

         $\hat{\theta}_i = (\hat{\gamma}_i, \hat{v}_i, \hat{\alpha}_i, \hat{\beta}_i) = f^{(k)}(\mathbf{x}_i)$

         $\hat{\mathbf{y}}_i = \hat{\gamma}_i$      // Infer a new set of pseudo-labels

         $\hat{\mathcal{U}}_i \leftarrow (\mathbf{x}_i, \hat{\mathbf{y}}_i)$      // Update pseudo-labels

         $\hat{\mathcal{W}}_i \leftarrow \mathrm{Var}[\mu]_i^{-1} = \hat{v}_i * (\hat{\alpha}_i - 1)/\hat{\beta}_i$     // Update adaptive weights

     **end**

**end**

---

and we want to reduce its effect on the model. Conversely, if a pseudo-label has low uncertainty, the label quality is high enough to be used as a proxy of a gold-label. Thus, we should assign a higher score in the loss for it. The uncertainty is from the teacher model in the previous episode and is fixed throughout the current episode. Thus, the adaptive weight for each pseudo data point $i$ becomes $\hat{\mathcal{W}}_i = \mathrm{Var}[\mu]_i^{-1}$. The final loss then becomes

$$\mathcal{L} = \frac{1}{|\mathcal{D}|} \sum_{i \in \mathcal{D}} \mathcal{L}_i^{\mathrm{evi}} + \sum_{i \in \mathcal{U}} \frac{\hat{\mathcal{W}}_i}{\sum_{i \in \mathcal{U}} \hat{\mathcal{W}}_i} \mathcal{L}_i^{\mathrm{evi}}, \tag{5}$$

where the first term corresponds to the labeled dataset $\mathcal{D}$ does not have weights unlike the second term corresponding to the unlabeled data. This adaptive loss solves two disadvantages since it has zero hyperparameters, and it removes the effect of bad pseudo-labels while retaining all training examples including the noisy ones to maximize the diversity of the training space.

## 5   PSEUD$\sigma$ MOTIVATION: CONNECTION TO ENTROPY MINIMIZATION

We derive motivation about why evidential uncertainty and the weighting mechanism could be beneficial to pseudo-labeling based on the entropy minimization framework for semi-supervised learning from Grandvalet & Bengio (2004; 2006); Lee et al. (2013). Notably, our use of Bayesian modeling enables us to analytically derive a conditional entropy for pseudo-labeled data. We find that evidential loss strongly relates to conditional entropy, and minimizing evidential loss directly minimizes entropy. Secondly, we find the conditional entropy could be decomposed into the inverse epistemic uncertainty and the log-likelihood, which motivates our weighting mechanism.

In the standard regression setting, one seeks to maximize the likelihood of the model $p_\theta(\mathcal{Y}|\mathcal{X})$ on the labeled data set $\mathcal{D}$. To utilize the unlabeled data set, we need to extract some useful information on how the model behaves on the unlabeled dataset and inject this information to improve the model. To measure the utility, entropy $\mathcal{H}(Y|\mathcal{U})$ is introduced (Grandvalet & Bengio, 2006) as a proxy to measure the amount of information in unlabeled data:

$$\mathcal{H}(\mathcal{Y}|\mathcal{U}) = \sum_{\mathbf{x}_i \in \mathcal{U}} \mathrm{E}_{y \sim \mathrm{p}_\theta(y \,|\, \mathbf{x}_i)}[-\log \mathrm{p}_\theta(y \,|\, \mathbf{x}_i)]. \tag{6}$$

Throughout the text, we are referring to entropy as Shannon entropy. Intuitively, high entropy is indicative of the poor model confidence on the unlabeled data point, while the low entropy can

be interpreted as high confidence of the model predictions. High entropy, associated with random predictions, while low entropy is associated with non-random behavior. Hence force, we hypothesize that small entropy may be indication of a signal that our model can benefit from. Small entropy, as seen below, corresponds to high model confidence. and vice versa. Large entropy corresponds to high model uncertainty. Entropy minimization framework casts the regression as the following optimization problem:

$$\arg\max_{\theta} \left[ \log p_{\theta}(\mathcal{Y}|\mathcal{X}) - c\,\mathcal{H}(\mathcal{Y}|\mathcal{U}) \right], \tag{7}$$

where $c$ is the proportionality constant. Intuitively, here, the objective tends to maximize the log-likelihood on the labeled dataset while minimizing the entropy on the unlabeled data set at the same time to transfer knowledge from unlabeled data.

In previous works (Schütt et al., 2017; Liu et al., 2021), molecule properties are not modeled probabilistically such that entropy calculation is infeasible. In contrast, PSEUD$\sigma$ uses Bayesian modeling approaches that allow us to analytically calculate the entropy. For every molecule $\mathbf{x}_i$ the machine learning model outputs four parameters $f(\mathbf{x}_i) = (\alpha_i, \beta_i, \gamma_i, \nu_i)$. Based on these parameters, the likelihood of label $y$ given the input molecule $\mathbf{x}_i$ is given by the Student's t-distribution in the context of evidential regression

$$p_{\theta}(y|\mathbf{x}_i) = \text{St}(y; \gamma_i, \sigma_{st,i}^2, 2\alpha_i) \tag{8}$$

evaluated at location parameter $\gamma_i$, Student's t-distribution scale parameter $\sigma_{st,i}^2 = \frac{\beta_i(1+\nu_i)}{\nu_i \alpha_i}$ and $2\alpha_i$ degrees of freedom. The entropy of the Student's t-distribution given in terms of evidential parameters is readily available (Appendix A):

$$\mathcal{H}(y\,|\,\mathbf{x}_i) = \frac{2\alpha_i+1}{2}\left(\Psi(\frac{2\alpha_i+1}{2}) - \Psi(\alpha_i)\right) + \log\sqrt{2\alpha_i}\,\text{B}(\alpha_i, \frac{1}{2}) + \frac{1}{2}\log\sigma_{st,i}^2\,, \tag{9}$$

where $\Psi$ is a digamma function and $\text{B}(\cdot\,,\cdot)$ is a beta function. If we take model (epistemic uncertainty) (Eq. 3) as our evidential uncertainty, we can show that minimizing entropy directly relates to minimizing epistemic uncertainty. We plot the relation between the entropy and epistemic uncertainty in the Figure 2.

As the next steps, we aim to uncover the dependence of the entropy on the model uncertainty of our pseudo-labeling approach. This can be done if we make two simplifications in entropy evaluation. Firstly, to introduce iterations as in pseudo-labeling, we replace entropy with the cross-entropy between two probability distributions: the predictions $y$ are generated from the probability distribution $p_{\theta(t-1)}(y|\mathbf{x}_i)$ at iteration step $t-1$ and log-likelihood are evaluated with respect to probability distribution $p_{\theta(t)}(y|\mathbf{x}_i)$ at iterative step t:

$$\mathcal{H}(\mathcal{Y}|\mathcal{U}) \approx \sum_{\mathbf{x}_i \in \mathcal{U}} \mathbb{E}_{y \sim p_{\theta(t-1)}(y|\mathbf{x}_i)}\left[-\log p_{\theta(t)}(y\,|\,\mathbf{x}_i)\right]. \tag{10}$$

Upon convergence, $t \to +\infty$, the probability distributions at every iterative step $p_{\theta(t-1)}(y|\mathbf{x}_i) \approx p_{\theta(t)}(y|\mathbf{x}_i)$ and are approximately the same and one can view introduced cross-entropy with respect to time step $t$ as entropy. At the earlier stages of training cross-entropy acts as a regularizer encouraging network parameters $\theta(t)$ to match $\theta(t-1)$.

As a second approximation, to uncover model uncertainty in mathematical formulas, we approximate the probability distribution at time step $t-1$. We resort to empirical estimate of the entropy, as done in (Grandvalet & Bengio, 2004). We select labels $y$ at the highest mode of probability distribution $p_{\theta(t-1)}(y\,|\,\mathbf{x}_i)$, which corresponds to $y = \gamma_i^{t-1}$. We obtain the following approximate for the entropy:

$$\mathcal{H}(\mathcal{Y}|\mathcal{U}) \approx \mathcal{H}_{emp}(\mathcal{Y}|\mathcal{U}) = -\sum_{\mathbf{x}_i \in \mathcal{U}} \mathcal{E}_i^{t-1}\,\log\,p_{\theta(t)}(\gamma_i^{t-1}|\mathbf{x}_i), \tag{11}$$

where the log probabilities are weighted by empirical probabilities as weights $\mathcal{E}_i^{t-1}$ evaluated at iterative step $t-1$ when plugged into Eq. 8 (also see Appendix. Eq. 18 for exact formula of Student's t-distribution)

$$\mathcal{E}_i^{t-1} = \text{St}(y = \gamma_i, \sigma_{st,i}^2, 2\alpha_i) = \frac{1}{\sqrt{2\,\alpha_i\,\sigma_{st,i}^2}\,\text{B}(\frac{1}{2}, \alpha_i)}. \tag{12}$$

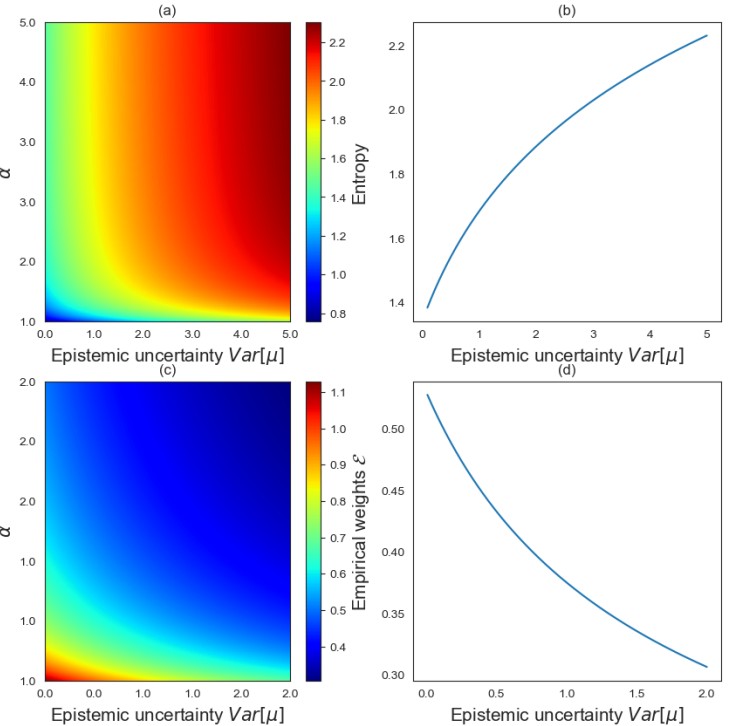

Figure 2: (a) Dependence of the entropy (Eq. 9) on epistemic uncertainty and virtual observation parameter $\alpha$ for a fixed aleatoric uncertainty $\mathbb{E}[\sigma^2] = 1$. As the the epistemic uncertainty increases, the entropy is also increases for all values of parameter $\alpha$. For example, figure (b) demonstrate the trend for a fixed $\alpha = 2$. Figure (c) demonstrates the dependence of empirical weights (Eq. 14) on epistemic uncertainty. The empirical weights tend to decrease as the epistemic uncertainty increases. Figure (d) demonstrates this trend for a fixed $\alpha = 2$.

To establish a relationship between empirical weights $\mathcal{E}_i^{t-1}$ and aleatoric $\mathbb{E}[\sigma_i^2]$ / epistemic $\text{Var}[\mu_i]$ uncertainties we rewrite

$$\sigma_{st,i}^2 = \frac{\alpha_i - 1}{\alpha_i} \left( \text{Var}[\mu_i] + \mathbb{E}[\sigma_i^2] \right) \tag{13}$$

$$\mathcal{E}_i^{t-1} = \frac{\left( \text{Var}[\mu_i] + \mathbb{E}[\sigma_i^2] \right)^{-\frac{1}{2}}}{\sqrt{2} \, \text{B}(\frac{1}{2}, \alpha_i) \sqrt{\alpha_i - 1}}. \tag{14}$$

Empirical coefficients depend on aleatoric, epistemic uncertainties and $\alpha_i$ parameter, which can be interpreted as virtual observations in support of the variance estimation (Jordan, 2009). In the limiting case $\alpha_i \gg 1$ one can approximate beta function via Stirling formula $\text{B}(\frac{1}{2}, \alpha_i) \approx \sqrt{\pi} \, \alpha_i^{-\frac{1}{2}}$ and empirical weights become

$$\mathcal{E}_i^{t-1} \approx \left( \text{Var}[\mu_i] + \mathbb{E}[\sigma_i^2] \right)^{-\frac{1}{2}}. \tag{15}$$

We can express the empirical coefficients depend both on aleatoric and epistemic uncertainties in a symmetric fashion.

We selected adaptive pseudo-labeling coefficients $\mathcal{W}_i$ in our pseudo-labeling approach Eq. 5 to be inverse epistemic/model uncertainties. We can see, that those coefficients directly relate to empirical coefficients derived from entropy minimization approach Eq. 14, as empirical coefficients also depend on model uncertainty in the inverse fashion. As the model uncertainty increases, the empirical coefficients $\mathcal{E}_i$ tend to decrease to minimize the entropy.

Table 1: Dataset statistics.

| Setting | Training Set | Validation Set | Testing Set | Unlabeled Set | OOD Set |
|---|---|---|---|---|---|
| Full-data | 110,000 (QM9) | 10,000 (QM9) | 10,831 (QM9) | 99,234 (PC9) | - |
| Low-data-1% | 1,100 (QM9) | 10,000 (QM9) | 10,831 (QM9) | 108,900 (QM9) | - |
| Low-data-10% | 11,000 (QM9) | 10,000 (QM9) | 10,831 (QM9) | 99,000 (QM9) | - |
| Out-of-distribution | 110,000 (QM9) | 10,000 (QM9) | 10,831 (QM9) | 99,234 (PC9) | 99,234 (PC9) |

Table 2: PSEUD$\sigma$ improves on full data setting. Reported metric is MAE. The lower the better.

| Property | Unit | SchNet | PhysNet | Cormorant | MGCN | DimeNet++ | SphereNet | PSEUD$\sigma$-S | PSEUD$\sigma$-D |
|---|---|---|---|---|---|---|---|---|---|
| $\epsilon_{\text{HOMO}}$ | meV | 41 | 32.9 | 36 | 42.1 | 24.6 | 23.6 | 32.9 | **20.4** |
| $\epsilon_{\text{LUMO}}$ | meV | 34 | 24.7 | 36 | 57.4 | 19.5 | 18.9 | 24.7 | **18.2** |

# 6 EXPERIMENTS

## 6.1 DATASET AND EXPERIMENTAL SETUPS

We evaluate PSEUD$\sigma$ using QM9 dataset (Wu et al., 2018) under two settings. *(A) Full-data*: We follow the previous works (Liu et al., 2021; Klicpera et al., 2020) where a 110,000/10,000/10,831 training/validation/testing set is obtained. For the unlabeled data, we solicit to PC9, a dataset of 99,234 molecules that consists of the same elements as QM9, curated by (Glavatskikh et al., 2019). *(B) Low-data*: we set $k\%$ of QM9 full training set as the training set (i.e. $k \times 110,000$) and we remove the label of $(1-k\%)$ QM9 full training set and make it as the unlabeled set. We evaluate in two $k$ values, 1 and 10, which means only 1,100/11,000 data points is trained respectively. Dataset statistics summary is in Table 1. Note that PC9 has a wider chemical diversity than QM9, demonstrated by wider distribution of distances of chemical bonds and more functional groups (Glavatskikh et al., 2019).

PSEUD$\sigma$ is model-agnostic. We evaluate it with two model backbones SchNet (Schütt et al., 2017) (PSEUD$\sigma$-S) and DimeNet++ (Klicpera et al., 2020) (PSEUD$\sigma$-D). We do not experiment with the SOTA atomistic model SphereNet (Liu et al., 2021) because it is highly computationally expensive. Our result is conducted on two targets $\sigma_{\text{HOMO}}, \sigma_{\text{LUMO}}$, because the PC9 dataset only has these two targets. We use mean absolute error as the evaluation metric.

For baselines, we compare with 6 state-of-the-art baselines, including SchNet (Schütt et al., 2017), PhysNet (Unke & Meuwly, 2019), Cormorant (Anderson et al., 2019), MGCN (Lu et al., 2019), DimeNet++ (Klicpera et al., 2020), and SphereNet (Liu et al., 2021). We report the best results taken from the original authors' paper while using the same fraction of data split in the full data setting. For PSEUD$\sigma$, we conduct two hyperparameter tunings on $\sigma_{\text{HOMO}}$ with SchNet backbone on the validation MAE with full data/low-data setting, respectively. The optimal hyperparameter is then used for both targets. Note that the atomistic model itself has the same hyperparameter as the original authors. Code will be released upon anonymous review period.

## 6.2 RESULTS

**Overview of results.** We report performances of PSEUD$\sigma$ in full data (Table 2), low-data (Table 3), out-of-distribution (Table 4) settings and find PSEUD$\sigma$ achieves the best performance across all settings, suggesting the robustness of the pseudo-labeling strategy. Systematic ablation study (Table 5) also show the importance of each module in PSEUD$\sigma$.

**PSEUD$\sigma$ improves on fully supervised QM calculations.** We report PSEUD$\sigma$ against 6 state-of-the-art models in Table 2. PSEUD$\sigma$-D surpasses all baselines in both targets $\sigma_{\text{HOMO}}, \sigma_{\text{LUMO}}$. Notably, PSEUD$\sigma$-D improves the SOTA by 3.2 meV, a significant margin. Particularly, comparing PSEUD$\sigma$-S with SchNet and PSEUD$\sigma$-D with DimeNet++, we find PSEUD$\sigma$ can consistently improve even on the fully supervised setting by a large margin (8.1 meV for SchNet and 4.2 meV for DimeNet++), highlighting the utility of PSEUD$\sigma$ and the high quality of PC9 as unlabeled data. It also shows that

Table 3: PSEUD$\sigma$ improves on low-data regime. Reported metric is MAE. The lower the better.

| Low-Data Setting | | 1% QM9 (1,100) | | 10% QM9 (11,000) | |
|---|---|---|---|---|---|
| Property | Unit | SchNet $\rightarrow$ PSEUD$\sigma$ | DimeNet++ $\rightarrow$ PSEUD$\sigma$ | SchNet $\rightarrow$ PSEUD$\sigma$ | DimeNet++ $\rightarrow$ PSEUD$\sigma$ |
| $\epsilon_{\text{HOMO}}$ | meV | 265.4 $\xrightarrow{+10.8}$ 276.2 | 248.9 $\xrightarrow{-18.7}$ 230.2 | 119.0 $\xrightarrow{-30.2}$ 88.8 | 81.1 $\xrightarrow{-13.7}$ 67.4 |
| $\epsilon_{\text{LUMO}}$ | meV | 290.6 $\xrightarrow{-57.8}$ 232.8 | 229.3 $\xrightarrow{-5.2}$ 224.1 | 93.3 $\xrightarrow{-15.0}$ 78.3 | 60.8 $\xrightarrow{-1.6}$ 59.2 |

Table 4: Out-of-distribution best validation MAE.

| Property | Unit | SchNet | DimeNet++ | PSEUD$\sigma$-D |
|---|---|---|---|---|
| $\sigma_{\text{HOMO}}$ | meV | 243.4 | 230.4 | **214.4** |
| $\sigma_{\text{LUMO}}$ | meV | 225.0 | 184.2 | **175.8** |

this direction of improving learning strategy instead of improving physics-based representation has potentials.

**PSEUD$\sigma$ significantly improves on low-data QM calculations.** In Table 3, we investigate how PSEUD$\sigma$ can improve on the low-data regime with only 1%, 10% training data point. That is to do QM using only 1,100 and 11,000 known data. This is to simulate the more expensive QM levels such as CCSD(T)/MP2. We observe PSEUD$\sigma$ can consistently and significantly improve the prediction in $\sigma_{\text{HOMO}}, \sigma_{\text{LUMO}}$ across both low-data settings and both model backbones, suggesting PSEUD$\sigma$ can help prediction in realistic low-data quantum calculations. Notably, in $\sigma_{\text{LUMO}}$ with 1% of QM9 data, PSEUD$\sigma$ improves upon SchNet by 57.8 meV, a considerable margin. We also observe that the gain margin is much more significant when the number of training data is smaller. This showcases the utility of PSEUD$\sigma$ in extremely low-resource settings, such as QM.

**PSEUD$\sigma$ improves out-of-distribution QM calculations.** Another realistic challenge is to infer accurately on unseen data distribution away from QM9. We conduct inference on the PC9 dataset where it has calculated $\sigma_{\text{HOMO}}, \sigma_{\text{LUMO}}$. We also find PSEUD$\sigma$ can again significantly improve OOD accuracy over DimeNet++, a SOTA method, with over 16.0 meV improvement on $\sigma_{\text{HOMO}}$ and 8.4 meV improvement on $\sigma_{\text{LUMO}}$, highlighting the robustness of PSEUD$\sigma$.

**Evidential uncertainty highly correlates to label quality.** PSEUD$\sigma$ utilizes uncertainty as a proxy of label quality because they are highly correlated for unseen molecules. In this experiment, we want to validate this hypothesis. We train on the complete QM9 training set with evidential uncertainty and then infer on the QM9 testing set. We find that the non-parametric Spearman correlation between MAE and epistemic uncertainty is 0.42 with a p-value < 1e-16. Additionally, we evaluate on PC9 out-of-distribution set, and the Spearman correlation is 0.35 with p-value < 1e-16, suggesting our uncertainty is a robust measure of label quality.

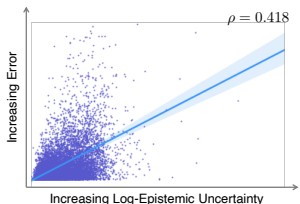

Figure 3: Uncertainty highly correlates to label quality.

**Ablations.** In Table 5, we conduct a systematic ablation study using SchNet as the backbone architecture on the fully supervised QM9 setting. We show that each component in PSEUD$\sigma$ is indispensable for PSEUD$\sigma$. In Table 2, we have reported original authors best performance following standard practices Klicpera et al. (2020); Liu et al. (2021). To further clearly demonstrate the utility of pseudo-labeling, in -pseudo-label, we keep all hyperparameters the same but remove the pseudo-labeling part. We show our pseudo-labeling strategy improve by a large margin. Next, in the -uncertainty ablation, we use a vanilla per-epoch pseudo-labeling strategy with no uncertainty, corresponding to a vanilla pseudo-labeling strategy. We demonstrate it does not work for QM as it even deteriorates compared to no pseudo-labeling (-pseudo-label), calling for specialized strategy design such as ours. Then, to compared with self-training strategy, -student ablation retrains a model in every episode as in (Xie et al., 2020) and the decreased performance shows that it is not ideal in QM calculations. Lastly, the -uniform ablation uses the same weight for all pseudo-labels with no uncertainty reweighting. The decreased performance shows the importance of detection and

Table 5: Ablation using SchNet as backbone on the fully supervised setting.

| Property | Unit | PSEUD$\sigma$ | -pseudo-label | -uncertainty | -student | -uniform |
|----------|------|------|------|------|------|------|
| $\epsilon_{\text{HOMO}}$ | meV | **32.9** | 38.9 | 47.7 | 41.4 | 37.2 |
| $\epsilon_{\text{LUMO}}$ | meV | **24.7** | 27.2 | 32.1 | 31.4 | 28.8 |

adaptive removal of bad pseudo-labels, achieved by our evidential characterization of the molecular property.

## 7 CONCLUSION

We introduce PSEUD$\sigma$, a simple, effective, model-agnostic pseudo-labeling strategy that can improve quantum calculations accuracy in abundant data, low data and out-of-distribution settings. PSEUD$\sigma$ learns from vast unlabeled data by assigning uncertainty-aware pseudo-labels. These pseudo-labels are adaptively selected to be absorbed into the model via an episodic schedule. Unlike eariler methods in QM that focuses on physics-based representation, we show the potential of data-centric approach.

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

## A  ENTROPY OF STUDENT'S T-DISTRIBUTION

While the entropy of the Student's t-distribution is well known, we derive it for completeness. Student's probability distribution defined in terms of location $\gamma$, scale factor $\sigma_{st}^2$ and $\nu_{st}$ degrees of freedom is

$$p(y; \gamma, \sigma_{st}^2, \nu_{st}) = \mathrm{St}(y; \gamma, \sigma_{st}^2, \nu_{st}) = \frac{\Gamma(\frac{\nu_{st}+1}{2})}{\sqrt{\nu_{st}\pi\sigma_{st}^2}\,\Gamma(\frac{\nu_{st}}{2})}\left(1 + \frac{1}{\nu_{st}}\frac{(y-\gamma)^2}{\sigma_{st}^2}\right)^{-\frac{\nu_{st}+1}{2}}, \quad (16)$$

where $\Gamma$ i a gamma function. Student's t-distribution can be written in terms of beta function $\mathrm{B} = \frac{\Gamma(x)\Gamma(y)}{\Gamma(x+y)}$ if we take advantage of the fact that $\Gamma(\frac{1}{2}) = \sqrt{\pi}$

$$\mathrm{p}(y; \sigma_{st}^2, \nu_{st}) = \mathrm{St}(y; \sigma_{st}, \nu_{st}) = \frac{1}{\sqrt{\nu_{st}\,\sigma_{st}^2}\,\mathrm{B}(\frac{1}{2}, \frac{\nu_{st}}{2})}\left(1 + \frac{1}{\nu_{st}}\frac{(y-\gamma)^2}{\sigma_{st}^2}\right)^{-\frac{\nu_{st}+1}{2}}. \quad (17)$$

In the main text of the manuscript, we used empirical estimate of Student's t-distribution, which corresponds to evaluation at the highest mode $y = \gamma$. Empirical estimation of probability becomes:

$$\mathrm{p}^{\mathrm{emp}}(y = \gamma; \sigma_{st}^2, \nu_{st}) \approx \mathrm{St}(y = \gamma; \sigma_{st}, \nu_{st}) = \frac{1}{\sqrt{\nu_{st}\,\sigma_{st}^2}\,\mathrm{B}(\frac{1}{2}, \frac{\nu_{st}}{2})}. \quad (18)$$

If we introduce a new variable $t = \frac{y-\gamma}{\sigma_{st}}$, Student's t-distribution converts into the standard form with probability density

$$\mathrm{p}(t; \nu_{st}) = \mathrm{St}(t; \nu_{st}) = \frac{1}{\sqrt{\nu_{st}}\,\mathrm{B}(\frac{1}{2}, \frac{\nu_{st}}{2})}\left(1 + \frac{t^2}{\nu_{st}}\right)^{-\frac{\nu_{st}+1}{2}}. \quad (19)$$

### A.1  PROPOSITION

**Proposition**: Entropy of the generalized and standard Student's t-distributions are related via the formula

$$\mathcal{H}(y; \sigma_{st}^2, \nu_{st}) = \mathcal{H}(t; \nu_{st}) + \frac{1}{2}\log\sigma_{st}^2. \quad (20)$$

**Proof:** The transformation $t = \mathrm{g}(y) = \frac{y-\gamma}{\sigma_{st}}$ is bijective and invertible with the inverse transformation $y = g^{-1}(t) = \sigma_{st}\,t + \gamma$. The Jacobin of the transformation g is $\frac{\mathrm{d}}{\mathrm{d}y}\,g(y) = \frac{1}{\sigma_{st}}$. According to the change of variables probability density formula

$$\mathrm{p}_y\,(y; \sigma_{st}^2, \nu_{st}) = \mathrm{p}_t\,(g(y); \nu_{st})\,|\,\frac{\mathrm{d}}{\mathrm{d}y}\,g(y)\,|. \quad (21)$$

The equation for the entropy transformation (equation 20) follows directly from the definition of the entropy.

To find the generalized entropy, we just need to calculate the entropy of the standard Student's t-distribution

$$\mathcal{H}(t; \nu_{st}) = -\int_{-\infty}^{+\infty} \mathrm{p}(t; \nu_{st})\log\mathrm{p}(t; \nu_{st})\,\mathrm{d}t = \quad (22)$$

$$\log\left(\sqrt{\nu_{st}}\,\mathrm{B}(\frac{1}{2}, \frac{\nu_{st}}{2})\right)\int_{-\infty}^{+\infty} \mathrm{p}(t; \nu_{st})\,\mathrm{d}t + \quad (23)$$

$$\frac{\nu_{st}+1}{2}\int_{-\infty}^{+\infty}\log(1 + \frac{t^2}{\nu_{st}})\,\mathrm{p}(t; \nu_{st})\,\mathrm{d}t = \log\left(\sqrt{\nu_{st}}\,\mathrm{B}(\frac{1}{2}, \frac{\nu_{st}}{2})\right) + \quad (24)$$

$$\frac{\nu_{st}+1}{2}\int_{-\infty}^{+\infty}\log(1 + \frac{t^2}{\nu_{st}})\,\mathrm{p}(t; \nu_{st})\,\mathrm{d}t. \quad (25)$$

To find the second integral, we make a substitution $x = \frac{t^2}{\nu_{st}}$ and obtain

$$\frac{\nu_{st}+1}{2} \int_{-\infty}^{+\infty} \log(1+\frac{t^2}{\nu_{st}}) \, \mathrm{p}(t;\nu_{st}) \, \mathrm{d}t = \frac{\nu_{st}+1}{2\,\mathrm{B}(\frac{1}{2},\frac{\nu_{st}}{2})} \int_0^{+\infty} \log(1+x)\,(1+x)^{-\frac{\nu_{st}+1}{2}}\frac{\mathrm{d}x}{\sqrt{x}} = \tag{26}$$

$$-\frac{\nu_{st}+1}{\mathrm{B}(\frac{1}{2},\frac{\nu_{st}}{2})}\frac{\partial}{\partial\nu_{st}} \int_0^{+\infty} (1+x)^{-\frac{\nu_{st}+1}{2}}\frac{\mathrm{d}x}{\sqrt{x}} = -\frac{\nu_{st}+1}{\mathrm{B}(\frac{1}{2},\frac{\nu_{st}}{2})}\frac{\partial}{\partial\nu_{st}} \int_0^{1} x^{\frac{\nu_{st}}{2}-1}\,(1-x)^{\frac{1}{2}-1} \, \mathrm{d}x = \tag{27}$$

$$-\frac{\nu_{st}+1}{\mathrm{B}(\frac{1}{2},\frac{\nu_{st}}{2})}\frac{\partial}{\partial\nu_{st}} \, \mathrm{B}(\frac{1}{2},\frac{\nu_{st}}{2}) = -(\nu_{st}+1)\frac{\partial}{\partial\nu_{st}} \log \mathrm{B}(\frac{1}{2},\frac{\nu_{st}}{2}) = \tag{28}$$

$$-(\nu_{st}+1)\frac{\partial}{\partial\nu_{st}} \left(\log\Gamma(\frac{\nu_{st}}{2}) - \log\Gamma(\frac{\nu_{st}+1}{2})\right) = \frac{\nu_{st}+1}{2} \left(\Psi(\frac{\nu_{st}+1}{2}) - \Psi(\frac{\nu_{st}}{2})\right), \tag{29}$$

where digamma function is defined as $\Psi(x) = \frac{\Gamma'(x)}{\Gamma(x)}$. Putting all the terms together, the entropy of the standard Student's t-distribution becomes

$$\mathcal{H}(t;\nu_{st}) = \frac{\nu_{st}+1}{2} \left(\Psi(\frac{\nu_{st}+1}{2}) - \Psi(\frac{\nu_{st}}{2})\right) + \log\left(\sqrt{\nu_{st}}\,\mathrm{B}(\frac{1}{2},\frac{\nu_{st}}{2})\right). \tag{30}$$

The final formula for the entropy of the labels $y$ is given by

$$\mathcal{H}(y;\sigma_{st}^2,\nu_{st}) = \frac{\nu_{st}+1}{2} \left(\Psi(\frac{\nu_{st}+1}{2}) - \Psi(\frac{\nu_{st}}{2})\right) + \log\left(\sqrt{\nu_{st}}\,\mathrm{B}(\frac{1}{2},\frac{\nu_{st}}{2})\right) + \frac{1}{2}\log\sigma_{st}^2. \tag{31}$$

## B  PSEUDO-LABELING, ENTROPY MINIMIZATION AND ALEATORIC UNCERTAINTY

In the main section of the text, we have considered the case where observed targets $(y_1,\cdots,y_i) \sim \mathcal{N}(\mu,\sigma^2)$ are drawn from the Normal distribution with unknown mean and variance $\mu$ and $\sigma^2$ and we have imposed a prior on them. The problem is significantly simplified if we treat $\mu$ and $\sigma^2$ in a deterministic way, such that our model $f$ outputs two parameters $\mu$ and $\sigma^2$. The model here is able to estimate aleatoric (data) uncertainty $\sigma^2$ but unable to model epistemic (model) uncertainty. By minimizing negative log-likelihood, the loss is significantly simpler than in Eq. 4.

$$\mathcal{L}_i = -\log\mathcal{N}(y_i;\mu_i,\sigma_i^2) = \frac{2\pi\sigma_i^2}{2} + \frac{(y_i-\mu_i)^2}{2\sigma_i^2}. \tag{32}$$

Empirical estimate of the entropy on the unlabeled data set becomes

$$\mathcal{H}(\mathcal{Y}|\mathcal{U}) = \sum_{\mathbf{x}_i\in\mathcal{U}} \mathrm{E}_{y\sim\mathrm{p}_\theta(y\,|\,\mathbf{x}_i)}[-\log\mathrm{p}_\theta(y\,|\,\mathbf{x}_i)] \approx -\sum_{\mathbf{x}_i\in\mathcal{U}} \mathcal{E}_i^{emp}\left[\log\mathrm{p}_\theta(y\,|\,\mathbf{x}_i)\right] \tag{33}$$

with log probability weights $\mathcal{E}_i^{emp} = \frac{1}{\sqrt{2\pi\sigma_i^2}}$. One can notice that the weights are inversely related to aleatoric uncertainties $\mathcal{E}_i \sim (\sigma_i^2)^{-\frac{1}{2}}$.

