# OpenReview forum: "Adaptive Pseudo-labeling for Quantum Calculations"
_ICLR.cc/2022/Conference — ICLR 2022 Submitted_

### Official Review · Reviewer_iJ3j · 2021-11-01

**Correctness:** 4
**Technical Novelty And Significance:** 3
**Empirical Novelty And Significance:** 4
**Recommendation:** 6
**Confidence:** 5

**Main Review:**

**Strong points**

- At first time, authors adapt and study pseudo-labeling training for quantum calculations problem which has its own challenges and motivations
- Authors propose to use evidential uncertainty to estimate uncertainty of the pseudo-labels and demonstrate that proposed adaptive weighting mechanism could be viewed as parts of the entropy minimization framework (or entropy minimization implicitly uses adaptive weighting)
- Demonstrating that algorithm works for different model backbones (SchNet and DimeNet++)
- In speech and image recognition it was demonstrated that the key components of iterative pseudo-labeling success are model noise (e.g. dropout) and data augmentation. This paper demonstrates that even with no noise (neither in data nor in model) pseudo-labeling can give significant performance boost.


**Weak points**
- Experiment for out of distribution testing is not enough. Ideal test would be evaluating trained models on some other dataset different from labeled and unlabeled data. One suggestion here (if no any other data is available) do the following: for low data 1% and 10% test also on PC9 (as it was not used in training at all) for all models; add one more experiment with 55k QM9 labeled + 55k rest QM9 as unlabeled + test on out of domain PC9.
- Would be nice to have also upper bound on out of distribution testing: finetune DimeNet++ baseline on PC9 labeled data with evaluating on QM9 and on some (not used) portion of PC9 - this is upper bound on what we can get with QM9 labeled and PC9 unlabeled.
- Fair “-uncertainty” ablation should follow exactly the same training except the weighting data, so that regenerating pseudo-labels should happen after N epochs. (Current “-uncertainty” ablation is proving another point that we should have some delay and enough training before regenerating pseudo-labels. However,  N is hyperparameter which I guess depends on the problem and data amount).
- Absense of ablation for low-resource setting as it could give entirely different dynamics of training compared to high-resource (at least this was observed in speech recognition).

**Related work**
- In speech recognition recently there were developed pseudo-labeled algorithms which continuously trains one model with regenerating pseudo-labels after several epochs https://arxiv.org/abs/2005.09267 and moreover using dynamic cache with pseudo-labels generated by some history model states https://arxiv.org/abs/2010.11524. These works are relevant in the way how authors performs iterative training with continuous model training. Moreover, the latter work (https://arxiv.org/abs/2010.11524) focuses on the low-resource setting of labeled data and solves problem of stable model training in low-resource. Another relevant work, also in speech recognition, is https://arxiv.org/abs/2106.08922: it uses exponential moving averaging of the model to predict pseudo-labels.
- Other relevant works https://arxiv.org/pdf/1908.02983.pdf, https://openreview.net/forum?id=SJgdnAVKDH
- It is worth to mention works related to modeling labels probabilistically / Bayesian modeling

**Comments**
- Algorithm 1: it would be clear to remove index $i$ for $f$, put $f^{k} = f^{(k-1)}$ in the beginning of the inner loop, simply put $Update(f^{k}, L)$. In any case Algorithm 1 needs improvements as right now indices are not clear.
- Noise strategy is not only on data side - it could be enough to have noise on model side via dropout, stochastic depths, etc. It is worth to mention that models for QM doesn’t not use these widespread techniques, so that main challenge is to introduce some noise into the training process. The latter is done is some extent with introduced uncertainty used to weight samples. Is there any possibility to introduce some noise in the model itself? One of the way is to try to use dynamic cache as in https://arxiv.org/abs/2010.11524 or EMA as in  https://arxiv.org/abs/2106.08922.
- In Figure 1 better to use "$K$ epochs" as $N$ is reserved for $N$ molecules
- "In the standard pseudo-labeling (Lee et al., 2013; Rizve et al., 2021), a set of labels are regenerated in every epoch." In (Lee et al., 2013) it is done every update, not epoch and the same model is continuously trained while in (Rizve et al., 2021) it is done after full model training, so some $K$ epochs of training, then new pseudo-labels are generated, model is reinitialized and trained from scratch on these new pseudo-labels.
- Did authors check this statement in their experiments? "Our key observation is that low-quality pseudo-labels have high model uncertainty, and high-quality pseudo-labels have low model uncertainty."
- "it removes a large set of unlabeled data, which reduces the diversity of the training space, harming the generalization." I do not agree with this statement: In (Rizve et al., 2021) the whole training continues until there will be no uncertain sample anymore, so in general at first pseudo-labeling rounds small portion of unlabeled data is used while at later rounds more data are used as model becomes stronger and more certain, thus diversity and generalization can be still hold. Maybe it is worth to smooth authors' formulation in the way that i) we can learn from noisy data starting right from the begging to extract necessary representations even if model is uncertain.
- How do authors weight supervised and unsupervised data? Or it is done via Eq. (5) which means use the proportion as it is in the original datasets? Do authors randomly sample batch from joint set of supervised and unsupervised data?
- Appendix: the Eq. (22) the right hand side should have additional 2 multiplier as variable change $x=t/\nu_{st}$ is done incorrectly (first you need to write left side as $\int_{-\inf}^{+\inf} ... = 2*\int_0^{+\inf} ...$ and then do variable change. This multiplier 2 should be propagated further till formula (28). Next, (26) and (27) are repetitions, so authors can skip (27). Everything else in Appendix is correct.
- Is there any connection between QM and assumption that $y_i\sim N(\mu, \sigma^2)$?
- To recheck: in Table 2 Pseudo$\sigma$-S has the exact same results as PhysNet. Is it correct?
- Any idea why for 1% train data for HOMO pseudo-labeling doesn’t work with SchNet? Does model diverge as soon as you start involving pseudo-labeled data?
- Could authors add a small paragraph in experiments on what is the difference between QM9 and PC9 data?
- Would be great to see the plot of average error and average uncertainty for different episodes, to demonstrate that both error and uncertainty are decreasing over the episodes.
- Why in Table 5 “-pseudo-label” is not the same as baseline SchNet in Table 2? If authors did optimization of hyperparameters for pseud$\sigma$ and we see that found hyperparameters even better for baselines - I think it is necessary to put retrained baselines also in Tables to have fair comparison of pseudo-labeling on top of baseline. Ideal case is that I have some baseline and following the paper simply add pseudo-labeling on top and I don’t need to redo optimization of the model hyperparameters again, and only deal with pseudo-labeling hyperparamenters).
- Does “-uncertainty” ablation diverge actually? Could authors show the training/validation curves? (as quality is worth that even baseline)
- For “-student” do authors train every episode till full convergence and then regenerate pseudo-labels and retrain from scratch?
- Please specify that ablations are done on full QM9 (this is obvious from MAE numbers, but would be good to have in the main text).
- When are first pseudo-labels generated? As model fully converged on supervised data?
- "This is important because we observe that per-epoch pseudo-labeling is far from converging, preventing the model from learning from all useful information in pseudo-labels in one episode.” Worse behaviour with per-epoch pseudo-labeling could be due to absence of any data/model augmentation which makes model to be more close to the model state with which pseudo-labels were generated after one epoch. This prevents from training. For future work, I would suggest authors to have a look on the ways to introduce some shift in the model (EMA or dynamic cache, etc.) which can help for convergence and bootstrapping from the model itself.
- Could authors provide more details on “For each episode, we also reinitialize the learning rate with a small step-wise decay strategy to allow the model a chance to jump out of the local optimum from the previous set of pseudo-labels.” Is it used in ablations too? Could authors report results without this trick? Is it used for both low-resource and high-resource experiments?

**Typos**
- Abstract typo: "pseud-labels" -> "pseudo-labels"
- Would be good for reader not familiar to quantum mechanics to specify abbreviations, e.g. CCSD(T) and MP2, also link to SchNet.
- Typo "in QM9 to test and validation model performance" -> "in QM9 to test and validate model performance"
- typo "at episode k then conduct inference" -> "at episode k then conducts inference"
- "$f (k + 1)$ after N epochs" -> "$f^{(k + 1)}$ after N epochs”
- "Dependence of the variance for the distribution of labels and variance for the parameter” - not clear English.
Eq (11) the right hand side: should it be $y_i^{t-1}$ -> $\gamma_i^{t-1}$
Eq. (9) $\sigma_i$ -> $\sigma_{st,i}$


**Summary Of The Paper:**

Self-training, or pseudo-labeling, is very simple and popular approach in different domains like speech and image recognition.
For the first time, pseudo-labeling is designed for quantum mechanics (QM) calculations problem. This problem is challenging for pseudo-labeling as it cannot use data augmentation ("small perturbational noise in 3D molecular geometry could easily lead to a drastic energy difference") or model noise (like dropout or layer drop due to design of models to work with 3D molecular geometry) which were found to be crucial for pseudo-labeling in other domains. Moreover, the main real case scenario for QM is small amount of supervised data available (low-resource) which makes pseudo-labeling is even harder. In this paper authors propose learning a single model with time-to-time regenerating pseudo-labels and incorporating these pseudo-labeled data according to the model uncertainty. The uncertainty is modeled in the following way: labels are assumed to be from normal distribution and model is designed to predict parameters of this normal distribution, thus model estimates expectation of the label and its dispersion. This uncertainty detects and prevents bad pseudo-label from affecting the model training by adaptively weighting the data according to inverse of uncertainty value. Pseudo-labeling results show that direction of improving learning strategy instead of improving physics-based representation has potentials. With experiments it is demonstrated that proposed pseudo-labeling algorithm, pseud$\sigma$, improves results over supervised baseline, works across different backbone models and applicable for low-resource regime. Moreover it demonstrates improved results for out of distribution data.

**Summary Of The Review:**

Overall authors uses classic semi-supervised algorithm, pseudo-labeling, but they propose necessary modifications like specific learning schedule, similar idea from other domains on bootstrapping a single model, and, the most important, regularizing training with uncertainty estimation for pseudo-labels and proper accounting it during optimization. All these modifications allowed pseudo-labeling to succeed to improve results for quantum mechanics (QM) calculations across different models, labeled data size and out of distribution data. All these modifications make a novel version of pseudo-labeling which is applied to the QM domain at first time and specifically designed for QM. Having all this in mind I recommend to accept the paper.

---

> ### Author Response · Authors · 2021-11-22
> **Response to Reviewer iJ3j**
>
> We greatly appreciate the reviewer’s time for providing detailed and constructive feedback. We believe the reviewer's comments have drastically improved our paper. We have conducted additional experiments and edited the paper extensively. Please see the updated paper. Thank you.
>
> RE: More experiments on OOD testing
>
> We thank the reviewer for the great suggestion. We have conducted an experiment on 55k QM9 labelled + 55k rest QM9 as unlabeled + test on out of domain PC9 with HOMO. Here we show the pattern on SchNet since DimeNet requires a much longer training time and it exceeds the rebuttal time. In the final version, we will conduct a full study on both SchNet and DimeNet across HOMO and LUMO. Here is the preliminary result:
>
> ---MAE on HOMO---
>
> SchNet: 267.7 mEV
>
> SchNet-D: 250.2 mEV
>
> ---End of Table---
>
> RE: Upper bound on OOD testing
>
> We thank the reviewer for another great suggestion. We train DimeNet++ baseline on 80%, validate on 10%, and test on 10% of PC9 labeled data. The MAE is 17.9 meV. Thus, the upper bound is high, leaving plenty of space for future improvements on OOD inference.
>
> RE: Clarification on the -uncertainty ablation
>
> We agree with the reviewer that it is crucial to have ablation on the weighting mechanism, which corresponds to the ‘-uniform’ ablation. We have clarified the distinction further in the updated paper.
>
> RE: Ablation in low-resource setting
>
> We thank the reviewer for this great suggestion. Due to the limited timeframe of the rebuttal, we decide to prioritize on the other experiments and are not able to conduct the full ablation for low resource setting at this time. But we will surely include them in the next version of the paper.
>
> RE: Related works
> We thank the reviewers for providing the additional references. We have added them to the related works section.
>
> RE: Comment
>
> We greatly appreciate the detailed and constructive comments. We have incorporated most of the comments in the updated paper. Here are the responses following the order of the comments:
>
> 1. We have greatly revised the algorithm 1 following reviewer 1’s suggestion and the indices are now clear.
>
> 2. Adding model noise sounds like a promising direction. One potential concern is that since the energy landscape is complex and sensitive to small perturbation, any form of the noise could introduce biased signals. However, we haven’t experimented with this, and we refer to future works.
>
> 3. We have updated the figure to replace N with K.
>
> 4. We have modified the Sec 4 Part 1 to reflect the proper usage of Lee et al and Rizve et al.
>
> 5. We have shown it in Figure 3 where the error is highly correlated to epistemic uncertainty in the testing set.
>
> 6. We agree with the reviewer for this key observation. We have modified the claim in the paper.
>
> 7. Yes, we randomly sample a batch from the joint set of supervised and unsupervised data. Note that the loss separates them apart.
>
> 8. Multiplier 2 is fixed and equation is updated
>
> 9. Normal is taken as the easy way to derive analytics solutions. However, indeed a rough distribution of QM9 energy is of a normal-shape.
>
> 10. Yes, this is due to a coincidence and they are different in the further decimals.
>
> 11. We do not have a good justification for this phenomenon. The model does not diverge but converge to a bad optimum. We suspect one potential driver is that the adaptive weighting is sensitive to outlier weights. In the learning dynamics, if a set of pseudo-labels set out a model in the wrong direction, it could reach a bad local optimum.
>
> 12. We have added a paragraph in the dataset description section and add reference to the original PC9 authors who conduct detailed chemical analysis. Notably, PC9 has wider bond distances distribution for several bond types and more functional groups than QM9, meaning a higher diversity.
>
> 13. We believe this is a good point. We will add it in the newer version of the paper.
>
> 14. The -pseudo-label is using the exact same hyperparameters as Pseudo, thus there is a minor difference from the SchNet in the original paper. But you can see that it strongly improves performance consistently. We agree with the reviewer and we plan to add a reproduced model of SchNet and DimeNet++  in Table 2.
>
> 15. The -uncertainty ablation does not diverge. It is simply reaching a bad local optimum. We will add it in the supplementary materials.
>
> 16. Yes, we use the same number of epochs $K$ because it is set to show convergence.
>
> 17. We have added texts about the ablation dataset in Table 5 and also in the ablation result section.
>
> 18. Yes, the first set of pseudo-labels are generated on a model trained only on supervised data.
>
> 19. We thank the reviewer for pointing out this future work.
>
> 20. Yes, this trick is used throughout the ablation. Empirically, we observe there is a slight advantage and we would include the exact number in the next version as an ablation study.
>
> RE: Typos
>
> We thank reviewer for catching these typos, which are fixed.

---

> > ### Comment · Reviewer_iJ3j · 2021-11-28
> > **Additional comments**
> >
> > Dear authors,
> >
> > Thanks a lot for detailed explanation, updated version of the paper and additional experiments I asked about. Please find below some discussion and comments regarding your updated paper and last comments.
> >
> > > Yes, we randomly sample a batch from the joint set of supervised and unsupervised data. Note that the loss separates them apart.
> >
> > This should be stated clearly in the paper, as weighting influences training, especially in case of low-resource setting. And with respect to loss formula (5) I expect that data are weighted to 1:1, so in batch half is sup. and half is unsup. (with pseudo-labels) data.
> >
> > Several additional comments:
> > - Authors need to include https://arxiv.org/abs/2005.09267 (Interspeech 2020) into Episodic Pseudo-labeling paragraph, as this work is not only related but exactly episodic training. The difference with the authors’ paper is that this prior work relies on data augmentation and language model incorporation (specific of speech recognition). I prefer to see that authors here state that they do not have any augmentations and other type of noise except weighting the pseudo-labels. I think this makes the main contribution of the paper that pseudo-labeling can work even in no standard noise (which we use now on daily basis) regime.
> > - Do authors use adaptive optimizers? If yes, then do they reset momentums too when learning rate is reinitialized?
> > - Is it meaningful from QM perspective to have energy prediction with >150 MAE for example (in the sense that current out-of-distribution results are applicable and useful)?
> > - I think the following things given in authors’ comments should be added directly into the text as they are important:
> >   - how the batch is formed
> >   - QM9 energy is roughly normal, so modeling with normal pdf is not only allows analytic solution but also is dictated by the physics.
> >   - State that any noise, data or model is not available, or at least we need carefully to design it in future (task is complicated because of physics)

---

### Official Review · Reviewer_zDAo · 2021-11-02

**Correctness:** 4
**Technical Novelty And Significance:** 2
**Empirical Novelty And Significance:** 3
**Recommendation:** 5
**Confidence:** 3

**Main Review:**

Strengths:
- Empirical results show improvements over competing methods
- Connection between evidential uncertainty and entropy minimization is interesting

Weaknesses:
- Contribution of the work is not clearly stated
- The technical novelty of the method is very limited: 1) Instead of regenerating pseudo-labels every epoch, they are regenerated less often (after an "episode": a set number of epochs). 2) Weighing the pseudo-label contribution to training loss by its uncertainty


Minor comments/edits:
- literature review (e.g. 2nd paragraph) needs references
- "concurrent work (Rizve et al., 2021)" A method from ICLR'21 is hardly "concurrent work".

**Summary Of The Paper:**

The paper is focused on improving quantum-mechanical properties of molecules in a setting similar to semi-supervised learning, where X is known for samples without Y. In an iterative fashion, the current model is used to predict pseudo-labels for those samples, then pseudo-labels together with labeled samples are used to improve the model. The paper uses existing approaches to quantify the uncertainty in individual pseudo-labels and then uses the uncertainty to weight their contribution to the loss.

**Summary Of The Review:**

The paper provides empirical evidence of improvement over competing methods, but the method is technically not very innovative.

---

> ### Author Response · Authors · 2021-11-22
> **Response to Reviewer zDAo**
>
> We thank the reviewer for taking the time to review our article. Below are our responses.
>
> RE: Contribution and novelty
>
> We thank the reviewer for the comment. Our novelty and contribution have six folds:
>
> (1) Previous QM focus on in-distribution and label abundant setting while we point out that the realistic ML-aided QM challenges lie in low-data and out-of-distribution settings.
>
> (2) Pivoting away from the status quo in improving the physics-based representation, we propose to look at data-centric approaches on learning from the vast array of unlabeled molecules.
>
> (3) Vanilla pseudo-labeling strategies that are important for other domains do not work well in QM. For example, data augmentation or model noise does not work since small perturbational noise in 3D molecular geometry could easily lead to a drastic energy difference. Also, we show that pseudo-label scheduling is crucial in QM since the continuous update or the student retraining after every set of pseudo-labeling do not work well in QM. Instead, we propose an episodic, non-student, non-stop, no-noise pseudo-label schedule that works well for QM9.
>
> (4) To prevent bad pseudo-label in biasing the training, we devise an adaptive weighting scheme where the weights are generated using evidential uncertainty. This is different from SOTA (Rizve et al., 2021) where a hyperparameter threshold is selected.
>
> (5) We derive a theoretical connection between the evidential loss and entropy minimization framework.
>
> (6) Empirically, we show we can improve QM accuracy for any atomistic model across full-data, low-data, and out-of-distribution settings.
>
> We have refurbished the last paragraph of introduction to explicitly point out the contribution and novelty of our work.
>
> Please also see reviews from reviewer iJ3j, which summarizes well about our contribution.
>
>
> RE: Difference with Rizve et al. 2021
>
> Our work is different from Rizve et al. 2021 in the following three ways:
>
> (1) Rizve et al. uses a fixed threshold to filter pseudo-labels, which introduces an additional hyperparameter - the threshold and also removes a large number of training samples at the beginning whereas our approach does not introduce new parameters and includes all noisy training samples.
>
> (2) Rizve et al. reinitialized the model after K epochs and train from scratch on a new set of pseudo-labels, whereas we continue to train (i.e. non-stop, non-student).
>
> (3) Rizve et al. look at the vision classification problem, and we look at QM regression, which introduces different uncertainty measures.
> We point out specific challenges in QM such as low-data and out-of-distribution learning and our algorithm is centered around QM.
>
> RE: More references in Section 2
>
> We have added additional related works. If the reviewer thinks any other reference is missing, please let us know, thanks!

---

### Official Review · Reviewer_5bAg · 2021-11-02

**Correctness:** 3
**Technical Novelty And Significance:** 4
**Empirical Novelty And Significance:** 4
**Recommendation:** 5
**Confidence:** 4

**Main Review:**

Strengths:

(1) The problem on QM is well-motivated and the paper is well written.

(2) The method part is well explained, even for readers like me who are not working on the self-training problems.

(3) From the technical point of view, this PseudSigma algorithm is interesting and promising. I expect it can be generalized to more general problems, in addition to the QM tasks.

(4) Almost consistent performance improvements can support PseudSigma.

-----

Weaknesses:

I mainly have two concerns (first two), and the remaining ones are just minor points.

(1) There is a gap in why using PseudSigma on QM tasks.
1. Why this assumption? First, the authors claim that PseudSigma is simple, effective, and model-agnostic (termed SEM for short). However, in Sec 4, the prior comes out abruptly. Similar for Sec 5 Eq 8, the Student-t distribution comes out abruptly. Can authors provide more intuitions of this assumption? And how is this formulation connected to SEM? One point I can imply from this paper is maybe due to the analytical solution? If so, authors should point this out explicitly. Otherwise, authors can provide different view points?
2. Why QM tasks? Both Sec 4 and Sec 5 are not related to this, i.e., the tasks here can be quite arbitrary. Thus, to better verify the effectiveness of PseudSigma, it is more reasonable to move to the standard self-training research line, with more standard self-training baselines. (This is the second point below.)

(2) Lack of self-training baselines.
Authors give a quite comprehensive introduction on self-training in Sec 2, and some are quite interesting, yet none of them is included in Sec 6. For example, `the perturbational noise in 3D conformation can lead to drastic energy differences`. This is a big difference between other applications and QM tasks, and can better reflect the importance of PseudSigma. The authors can consider adding it into baselines.


(3) In Sec 5, the authors claim that `molecule properties are not modeled probabilistically ...`, which I cannot fully agree with. So for regression QM tasks, MSE/MAE is indeed modeling the y under Gaussian/Laplacian distribution.

(4) In Sec 2, Pseudo-labeling paragraph, there are two redundant citations (Xie et al.) in one sentence. The authors can remove the second.

(5) In Sec 3, the input X also contains the atom types.


**Summary Of The Paper:**

This paper starts from two real challenges in QM problems: OOD and low-data issues. Then it proposes a self-training methods called PseudSigma. PseudSigma is a simple, effective, and model-agnostic algorithm with robust empirical improvements.

**Summary Of The Review:**

Both the strengths and weaknesses of this paper are obvious.

- From the technical point of view, the PseudSigma is promising.
- From the story telling point of view, it has some gaps, especially about how PseudSigma is connected to QM tasks.

I would give a borderline rate for now, and see what are authors response during reubttal.

---

> ### Author Response · Authors · 2021-11-22
> **Response to Reviewer 5bAg**
>
> We thank the reviewer for the positive feedback and constructive advice. The reviewer has raised great points, and below is our response to the comments.
>
> RE: Why assumptions of PseudoSigma?
>
> We thank the reviewer for raising this important point. The main framework of PseudSigma is simple - generate pseudo-labels in an episodic fashion where they are weighted by uncertainty. In practice, uncertainty requires Bayesian modeling of the label, thus we put priors to model the molecule property as it allows analytical solutions of epistemic uncertainty. However, the algorithm requires minimal changes to any molecule ML model, since it only needs to modify the loss function and the training loop. Also, the performance boost is strong with this minimal engineering in various scenarios. Thus, we claim that it is simple, effective and model-agnostic. We have updated the paper to make this point clear in Sec 2 part 2.
>
> Regarding Sec 5, we aim to connect the method with the entropy minimization framework, which introduces some technicalities but the method itself in practice is not affected by it and is thus simple and straightforward.
>
>
> RE: Why QM?
>
> QM is a very important application of ML, since it is crucial in drug discovery and could potentially save lots of time and cost, especially in the wake of COVID-19. However, the current focus on ML in QM is on improving the physics based representation (e.g. model architecture innovation) and we have seen a saturation of performance. Our paper argues that the community should take a pivot and look at a data-centric AI approach by learning from unlabeled data. Also, as the reviewer pointed out, classic tricks in self-training that work well in the vision domain (e.g. adding noise), do not work for QM. Thus, how to effectively design a self-training approach for QM is missing in the literature. Overall, based on these two reasons, we believe it is timely to study this important direction to improve ML-aided QM.
>
> RE: Perturbational Noise Baseline
>
> We thank the reviewer for this excellent point. Indeed, since QM energy is solely decided by the 3D coordinates, and is sensitive to small changes in XYZ, the classic noisy-student approach in self-training does not work. Following reviewer suggestion, we have added it as an ablation study where we added a gaussian noise for XYZ in pseudo-labeled samples and we see a performance drop, suggesting the importance of a customized self-training approach for QM.
>
> ---MAE on HOMO---
>
> SchNet+Pseudo: 32.9 mEV
>
> SchNet+Pseudo+Noise: 41.3 mEV
>
> ---End of Table---
>
> RE: Probabilistic modeling of molecular properties
>
> We thank the reviewer for this critical point. Although MSE/MAE implicitly use gaussian/laplacian, they do not generate a score that can be leveraged to measure epistemic uncertainty in order for us to weight the pseudo sample. To do that, we model the probability distribution of the label explicitly by outputting label distribution parameters which can directly calculate epistemic uncertainty. We further clarify in the updated paper.
>
> RE: Redundant citations
>
> We thank the reviewer for the great catch. We have removed the second citation.
>
> RE: Atom types notations
>
> We have added notations for additional features of the molecule.

---

### Official Review · Reviewer_ianT · 2021-11-03

**Correctness:** 3
**Technical Novelty And Significance:** 1
**Empirical Novelty And Significance:** 2
**Recommendation:** 3
**Confidence:** 3

**Main Review:**

**strengths**
* Label scarcity is an important problem when using ML/DL to train models for molecular property prediction, especially when the goal is to get to similar accuracies such as more expensive computational methods such as coupled cluster methods.
*The empirical results seem encouraging.

**weaknesses**
* The method is not explained well. The paper can be significantly improved in this aspect. For example:
	-  The paper contains pseudocode to explain the algorithm, but there are issues with notation, both in this algorithm and in the main text.
		* algorithm 1: isn't there a line missing where $f^{(k)}$ is set equal to $f^{(k-1)}$ ?
		* algorithm 1: In the inner loop over the dataset $\mathcal T$, the index i is used as a subscript to the update on the neural network function. However, this index is never updated and comes implicitly from the subscript of the datapoint.  I don't understand why the neural network function f needs a subscript here.
		* algorithm 1: In the inner loop over the dataset $\mathcal T$,  the $f^{(k-1)}$ model is used to generate predictions, uncertainties and it is updated. Shouldn't the uncertainties come from the teacher model that was used to generate the pseudo labels?
		* algorithm1: the distinction between the model that produces the pseudolabel and the model that optimizes a loss based on its own predictions and the pseudolabels is not clear.

	- The paragraph on "Evidential uncertainty quantification" on page 4 contains many confusing statements and notations.
		* just below eq 1 is the statement "dependence of the variance of the distribution of labels and variance for the parameter $\mu$ allows us to factorize the posterior for$\mu$ and $\sigma$ independently." But in eq 1 the normal distribution for $\mu$ depends on $\sigma$, so it is not factorized independently , but rather p(mu, sigma) = p(mu|sigma)p(sigma).
		* eq 2 shows the mean of  $\sigma^2$ as the mean of the inverse gamma distribution. However, in eq 1 $\sigma$ is distributed according to the inverse gamma distribution, not $\sigma^2$.
		* Where in algorithm 1  is the loss in eq 4 used?
	- Notation is inconsistent or unclear:
		* At end of section 3, page 3: f is used to denote the model that generates pseudo labels, as well as for the model that is trained using ground truth labels and pseudolabels. In other words it is used for both the teacher and student model. This gets very confusing for the reader.
         * The unlabeled dataset is defined as $\mathcal X_U$ at the end of section 3, but in section 4 it is denoted with $\mathcal U$. Datapoints in the unlabeled dataset are sometimes denoted with $x_i$ and sometimes with $\mathcal U_i$ and sometimes with $\mathcal X_i$.


* Section 5 attempts to derive a motivation for the weighting of datapoints depending on the confidence of the model's pseudolabels. However, I found this section very unclear.
	* I don't see how the middle plot in figure 2 shows that minimizing the evidential loss can minimize the entropy. A lower total evidence in that plot leads to a higher entropy.
	* why are $\beta=1$ and $\nu=1$ representative parameters in figure 2.? What is the motivation for this particular choice? How does the plot change if you change these values?
	* what does the y axis label on the right plot of figure 2,$p(y=y)$, mean?
	*  At the top of page 6 the authors state "intuitively, we show that minimizing evidential loss can implicitly minimize entropy on the unlabeled data such that it encourages positive information transfer." What is the intuition behind minimizing the entropy of the unlabeled data predictions. That would mean you want your model to become confident on unlabeled data? I don't see why that is desirable.
	* The discussion and derivation around eq 10, 11 and 12 is very confusing. For eq 10, I don't see what expectation maximization has to do with using a different distribution to sample from compared to the distribution that appears inside the expectation. I don't see how eq 11 is a valid empirical estimate of the expectation in eq 10.  eq. 11 is missing a minus sign. Where does the student distribution come from in eq 12?
   * top of page 6: states that the second term in eq 4 is the total evidence. The second term contains 2v+a. But in the x-asis of plot figure 2, we see 2v + a. which one is correct?

* It would be useful to comment on the difference between molecules described in QM9 and PC9, so the reader can get a better sense of how hard it is to generalize across these two datasets and how "out of distribution" molecules of PC9 are compared to those in the QM9 dataset.
* Comments on experiments:
    - what is the explanation for why using pseudolabels deteriorates performance on the $\epsilon_{homo}$ prediction for SchNet in table 3 for 1% labeled data?
    -  Although I appreciate the fact that the authors did an ablation study, several things are unclear. What is the difference between -pseudolabels and just a SchNet model?  The -uncertainty and -student baselines perform worse than the SchNet baseline (MAE 41 in table 2) on the $\epsilon_{homo}$ prediction. How come?

**Minor comments/questions**:
* CCSD and MP2 acronyms are not defined in introduction.
* In the introduction the acronym DFT is used before it is defined.
* SchNet is not accompanied by a reference at the end of paragraph 2 in the introduction.
* please explain what gold labels means early in the manuscript.

**Summary Of The Paper:**

This paper proposes to use pseudo-labeling/self-training for the prediction of molecular properties which are traditionally computed with quantum mechanical chemistry methods such as density functional theory or coupled cluster methods. While several recent works have trained supervised models on the QM9 dataset with labels produced by density function theory, labeled data for more accurate methods such as coupled cluster methods are significantly more expensive to obtain. This paper proposes to approach this label scarcity issue with pseudo-labeling/self-training on unlabeled data. The quality of the pseudo-labels is taken into account as follows: datapoints for which the model produces a pseudolabel with large uncertainty are downweighted compared to datapoints for which the model has high confidence on the pseudolabel.

The proposed training strategy is used to train models on different versions of QM9 and PC9. The first case considers training on all the labeled data in the training set of QM9, and the data of PC9 is used as unlabeled data. The second case considers only the QM9 dataset, where only 1% of the labels are used (and the rest is used as unlabeled data), or 10% of the labels are used.

Empirically the authors observe that using additional unlabeled data with pseudolabels improves the performance of the baseline models, even when only a small fraction of ground truth labels are available.
The authors check their hypothesis on the test set of QM9 that the model's uncertainty is well calibrated, in the sense that MAE and the epistemic uncertainty are positively correlated.


**Summary Of The Review:**

On the positive side, the topic is interesting and timely and I expect this could be of interest to the research community. Furthermore, the empirical results are encouraging. However, as explained above, the paper could be greatly improved in terms of explaining the method. Furthermore, some sections (such as section 5) contain problematic derivations and inconsistent notation. The ablation study is not explained well. At this point I think the weaknesses outweigh the positives, so I recommend rejection.

---

> ### Author Response · Authors · 2021-11-22
> **Response to ianT - Part 1**
>
> We thank the reviewer for the valuable comments and for the positive feedback on the topic, method, and empirical result. We acknowledge the lack of explanations and thus have extensively improved our presentation and conducted additional experiments in the updated manuscript, which, we believe, addressed all of your comments. All changes are marked in red. Please see the updated paper and the point-by-point response below and let us know if you have any other questions.
>
> RE: Clarification on the pseudo-code (algorithm 1)
>
> We thank the reviewer for the detailed comments. Below, we respond to each of the concerns following the order of comment:
>
> 1: We have added a line that sets $f^{(k)} = f^{(k-1)}.$
>
> 2: The subscript on $f$ was denoted as the index of step, which we agree with the reviewer could be removed to improve clarity.
>
> 3: In the inner loop, the evidential parameters are generated to update the evidential loss, while the weights to adjust pseudo-labels are from the teacher model in the previous episode. To clarify, we have modified algorithm 1 by adding explicitly $\mathcal{W}_i$ and also added an explanation paragraph before equation 5.
>
> 4. Pseudo-label is only generated after each episode and it is not trainable. Other than that, it could be considered as a training data point along with the gold-labelled data, where the evidential loss is optimized on top of them within each episode in the inner loop. We have added comments in algorithm 1 to further explain.
>
> RE: Clarification on evidential uncertainty quantification
>
> We thank the reviewer for raising these issues. Here, we respond to each of the concerns following the order of comment:
>
> 1. We modify the text to clarify that we assume the posterior can be factorized such that $p(\mu, \sigma^2) = p(\mu)p(\sigma^2)$ where the resulting posterior follows the NormalInvGamma distribution.
>
> 2. Thank you for spotting the typo, it is $\sigma^2$ that follows the inverse gamma distribution. We have modified it accordingly.
>
> 3. Loss in eq 4 is evidential loss for each data point and it is included in the eq 5 as $L^{evi}_i$. In algorithm 1, we make a simplification to only point to eq 5.
>
> RE: Clarification on notations
>
> We thank the reviewer for the detailed comment. Here are our responses following the order of comments:
>
> 1. Our model does not have a separate student and teacher model. In fact, in the ablation study, we show a separate student model in each episode is not improving on QM tasks. Instead, we did an episodic training strategy, where the model is constantly trained throughout all episodes, and each episode has a new and improved set of pseudo-labels. This new strategy allows the model to be exposed to a larger number of training data points given the same time frame. We have added a description in the end of page 3 section 3.
>
> 2. We have made edits throughout the paper to change the unlabeled set notation as $\mathcal{U}$ and each data point as $\mathbf{x}_i$.
>
> [Continue in the next comment...]

---

> > ### Author Response · Authors · 2021-11-22
> > **Response to ianT - Part 2**
> >
> > RE: Further explanation on theoretical motivations
> >
> > We thank the reviewer for the constructive feedback. Below, we resolve each of the concerns following the order of comment:
> >
> > 1. Lower total evidence means that the pseudo data point does not have much evidence to support it, thus the model is not confident about it, or is highly uncertain about it. In this plot, we show that our Bayesian modeling can establish an analytical positive relation between evidence and entropy. Since the evidential loss would encourage higher total evidence when the gap between ground truth and prediction is low, we see that it implicitly tries to minimize entropy for high-quality pseudo-labels.
> > 2. We have generated a 2d plot (for different parameters) to demonstrate the dependence of the entropy on model uncertainty. The pattern is consistent.
> > 3. We have updated the plot to make the visualization more clear, to demonstrate the dependence of weight coefficients on the model uncertainty.
> > 4. Our reasoning is based on the intuition that random behaviour (no signal) produces high entropy while low entropy is associated with non-random behaviour. Hence force, we hypothesize that small entropy may be an indication of a signal that our model can benefit from. As a result, small entropy corresponds to high model confidence.
> > 5. We made the revision to help clarify the simplifications that we made in the entropy evaluation. For eq 10, we renamed expectation maximization with cross-entropy evaluation and explained two probability distributions used in this evaluation and why in the limiting case t-> infinity this can be viewed as entropy. For eq 11, the empirical estimate is derived from [Grandvalet and Bengio, 2014]. In eq 11, the minus sign in the equation for the entropy is corrected. In eq 12, we use the Student’s t-distribution to model the likelihood (see eq 8). All the above-mentioned changes are extended and updated in the manuscript.
> > 6. We have removed the total evidence plot in figure 2 to simplify the notations. Note that total evidence is part of the second term in Eq 4, not the entire term. The second term is a regularizer that encourages lower total evidence, which correlates positively to higher epistemic uncertainty when the gap between ground truth and prediction is high. We have added further clarifications in the paragraph below Eq 4.
> >
> > RE: ​​On the difference between molecules described in QM9 and PC9
> >
> > We thank the reviewer for the valuable comments. [Glavatskikh et al. 2019] has conducted analysis on the difference between QM9 and PC9 and they find that PC9 contains a much higher chemical diversity and thus out-of-distribution of QM9: (1) bond distance analysis shows that C-F/N-N bond has wider distance in PC9, showing larger chemical diversity; (2) functional group analysis shows that PC9 has 97 and QM9 only has 71, further shows PC9 diversity. This distinction is made clear in Sec 6.1 and we also added a pointer to [Glavatskikh et al. 2019] for more information to the interested readers.
> >
> > RE: On ablation
> >
> > We thank the reviewer for the clarification question. Note that following the common practices ([Liu et al. SphereNet, Klicpera et al. DimeNet]) in QM benchmarking due to the extensive computational complexity, Table 2 contains the best performance obtained by the original authors. In Table 5, -pseudo-label is indeed the SchNet model, but it is with our implementation with the same parameters as Pseudo-S where the only difference is the pseudo-labeling part. Thus, you can see that our pseudo-labeling strategy really can improve performance by quite a large margin for SchNet. In addition, -uncertainty and -student indeed perform worse than the original model, highlighting that a simple application of pseudo-labeling does not work for QM and motivating the importance of our approach.
> >
> > RE: Minor comments
> >
> > We have expanded the acronym of CCSD/MP2/DFT, added references to SchNet and explained the gold label in the introduction.
> >
> > Summary of Response
> >
> > We have addressed all of the reviewer's comments point by point and greatly refurbished the paper. Let us know if you have any further questions. Thank you!

---

### Decision · Program_Chairs · 2022-01-20

**Decision:**

Reject

**Comment:**

The reviewers were split on this paper: the positive review appreciated (a) how adaptive weighing can be viewed as part of energy minimization, (b) the flexibility of the model to work with different model backbones, (c) the demonstration that even in no-noise settings the method generates noticeable improvements. However, all reviews saw important shortcomings in the (a) few out-of-distribution results, (b) limited ablation studies, (c) clarity of the writing, particularly in notation, (d) explanations of experimental results (e.g., why using pseudolabels sometimes deteriorates performance), (e) assumptions behind the proposed method, (f) lack of self-training baselines, (g) limited technical novelty. Ultimately, the number and severity of the shortcomings outweigh the positive parts of the paper. If the authors take the reviewer’s recommendations into account the paper will be a much stronger submission.